

# Spatial-temporal variation and impacts of drought in Xinjiang (Northwest China) during 1961–2015

Junqiang Yao[1], Yong Zhao[2] and Xiaojing Yu[1]

[1] Institute of Desert Meteorology, China Meteorological Administration, Urumqi, China
[2] School of Atmospheric Science, Chengdu University of Information Technology, Chengdu, China

## ABSTRACT

Observations indicate that temperature and precipitation patterns changed dramatically in Xinjiang, northwestern China, between 1961 and 2015. Dramatic changes in climatic conditions can bring about adverse effects. Specifically, meteorological drought severity based on the standardized precipitation index (SPI) and the standardized precipitation evapotranspiration index (SPEI) showed a decreasing trend in Xinjiang prior to 1997, after which the trend reversed. SPEI-based drought severity shows a much stronger change during 1997–2015 than the SPI, which is independent of the effect of evaporative demand. Meteorological drought severity has been aggravated by a significant rise in temperature (1.1 °C) over the last two decades that has not been accompanied by a corresponding increase in precipitation. As a result, the evaporative demand in Xinjiang has risen. An examination of a large spatio-temporal extent has made the aggravated drought conditions more evident. Our results indicate that increased meteorological drought severity has had a direct effect on the normalized difference vegetation index (NDVI) and river discharge. The NDVI exhibited a significant decrease during the period 1998–2013 compared to 1982–1997, a decrease that was found to be caused by increased soil moisture loss. A positive relationship was recorded between evaporative demand and the runoff coefficients of the 68 inland river catchments in northwestern China. In the future, meteorological drought severity will likely increase in arid and semiarid regions as global warming continues.

## INTRODUCTION

Severe droughts have occurred more frequently as a result of recent warming, especially in Africa and Asia (*Li, Chen & Yuan, 2015*; *Li et al., 2015a*; *Li et al., 2015b*; *Li et al., 2015c*; *Dai, 2011*; *Zhang & Zhou, 2015*). Drought is the most damaging, widespread, and urgent natural disaster, with negative impacts on agriculture, ecosystems, hydrology, economic and social activities, and environmental health (*Vicente-Serrano, Beguería & López-Moreno, 2010*; *Vicente-Serrano et al., 2010*; *Wang, Lettenmaier & Sheffield, 2011*; *Yu et al., 2014*; *Liu & Jiang, 2014*; *Chen & Sun, 2015*). Droughts are often grouped into four types: meteorological, hydrological, agricultural, and socioeconomic (*Council, 2004*; *Benitez & Domecq, 2014*;

Corresponding authors
Junqiang Yao, yaojq@idm.cn, yaojq1987@126.com
Yong Zhao, zhaoyong@idm.cn

*Mishra & Singh, 2011*). Meteorological drought precedes and induces other types of droughts, and local feedbacks often enhance atmospheric anomalies (*Trenberth, Branstator & Arkin, 1988*; *Dai, 2011*; *Potop et al., 2014*; *Li, Chen & Yuan, 2015*; *Li et al., 2015a*; *Li et al., 2015b*; *Li et al., 2015c*). A better understanding of drought is therefore of primary importance in order to enable early warnings of threats to environmental resources.

East Asia, which is dominated by the Pacific-Japan and Silk Road long-distance connection patterns (*Zhang & Zhou, 2015*), is highly influenced by drought. China is mainly located in East Asia and is strongly impacted by drought (*Zhang & Zhou, 2015*; *Li, Chen & Yuan, 2015*; *Li et al., 2015a*; *Li et al., 2015b*; *Li et al., 2015c*). Drought severity has increased significantly in the past half-century, causing substantial damage (*Xu et al., 2015a*; *Xu et al., 2015b*; *Wang et al., 2012*). Both *Zhang & Zhou (2015)* and *Yu et al. (2014)* report that severe drought has become more frequent since the late 1990s all over China, with particularly strong increases in southwestern and northern China (*Wu et al., 2011*; *Yang et al., 2013*). Encouragingly, however, drought severity has decreased in Xinjiang and on the Tibetan Plateau (*Wang, Lettenmaier & Sheffield, 2011*; *Zhang et al., 2012*), and the drought situation in the Huang-Huai-Hai Plain of China has remained stable for the past 30 years (*Wang et al., 2015*). *Xu et al. (2015a)* presented the spatio-temporal variation of drought in China based on the 3-dimensional clustering method.

In arid regions, widespread drying is common because precipitation depends strongly on a few precipitation events; if these events do not occur, the region becomes increasingly dry (*Dai, 2011*; *Sun et al., 2006*; *Li et al., 2017*). *Meza (2013)* reported significantly increased drought trends based on SPEI values over northern and central Chile (one of the driest regions in the arid Americas). In East Asia, most drought studies have concentrated on the eastern monsoon region although arid regions in China have sustained severe and costly droughts in the past few decades (*Wang et al., 2015*). For example, eastern arid China suffered a severe drought from 1994 to 1995 that caused a 32% decrease in summer grain production (*Zhang & Fang, 1995*). In northwestern China, drought-affected and drought-damaged area has increased over the past two decades, and the annual damaged area reached $0.36 \times 10^7$ ha in the 1990s (*Wang et al., 2015*). *Zhang et al. (2012)* found that droughts in Xinjiang have weakened, but that the change properties are different in different regions of Xinjiang. Using precipitation-based drought indices, *Li et al. (2016)* also indicated that drought severity has decreased. *Tao et al. (2014)* indicated that the frequency of moderate and severe droughts has decreased in the Tarim River basin, Xinjiang, but that extreme drought became slightly more prevalent after 1986. *Zhang et al. (2015)* assessed the vulnerability of the Tarim River basin to drought and found that drought dynamics can vary within a larger region and among indices.

Various indices have been developed to depict and monitor drought events more effectively. For example, the standardized precipitation index (SPI; *Svoboda, Hayes & Wood, 2012*) and the Palmer drought severity index (PDSI; *Palmer, 1965*) have been widely used in the meteorological community to monitor drought severity (Palmer, 1965; *McKee, Doesken & Kleist, 1993*; *Vicente-Serrano et al., 2011*). The PDSI is based on a primitive water balance that incorporates prior precipitation evaporation demand, moisture supply, runoff,

and soil water-holding capacity (*Vicente-Serrano, Beguería & López-Moreno, 2010*; *Vicente-Serrano et al., 2010*; *Zhang et al., 2015*; *Wang et al., 2015*). Despite its physical foundation and the inclusion of evaporative demand, it lacks multi-scalar characteristics. This is a critical weakness of PDSI because drought is a multi-scale phenomenon (*Vicente-Serrano et al., 2011*). In contrast, SPI has the advantage of quantifying drought on different time scales and has been used by the WMO as a reference drought index (*McKee, Doesken & Kleist, 1993*; *Hayes et al., 2011; Svoboda, Hayes & Wood, 2012*). However, SPI also has shortcomings in that it considers only precipitation data without using evaporative demand, thereby neglecting the effects of global warming (*Taylor et al., 2012*; *Teuling et al., 2013*; *Beguería et al., 2014*; *Cook et al., 2014*; *Zhang et al., 2015*; *Xu et al., 2015a*; *Xu et al., 2015b*).

More recently, the standardized precipitation evapotranspiration index (SPEI), developed by *Vicente-Serrano, Beguería & López-Moreno (2010)* and *Vicente-Serrano et al. (2010)*, takes into account the monthly climatic water balance (the difference between precipitation and reference evapotranspiration, P-PET) rather than using only precipitation. It combines the multi-scalar character of the SPI and the sensitivity of PDSI with changes in evaporative demand (*Wang et al., 2015*). *Vicente-Serrano et al. (2015)* provided the contribution of precipitation and evaporative demand to drought indices for different climates, and SPEI reflects the most evident sensitivity to PET, which is mainly controlled by aridity. *Vicente-Serrano et al. (2015)* also used the SPEI to reveal the increased drought severity caused by temperature rise in southern Europe. Hence, SPEI has proven to be an effective tool for monitoring and studying recent droughts under warming. Recently, it has been used in diverse studies that have analyzed drought variability, drought atmospheric mechanisms, and drought impacts over many parts of the world, including Asia, North America, Africa, Europe, and Australia (*Allen et al., 2011*; *Potop, 2011*; *Abiodun et al., 2013*; *Wang et al., 2015*; *Zhang et al., 2016*).

In arid regions, droughts result in significant water shortages that have critical repercussions on socioeconomic development (*Zhang et al., 2012*; *Sun & Ma, 2015*). The dynamics of droughts may become more sensitive under climate change (*Jenkins & Warren, 2015*), and the risk of drought has increased as a result of the global warming that has occurred over the past half-century (*Zhang et al., 2016*). Arid regions in northwestern China now face severe drought risk and water crises. Obtaining a better understanding of drought variation in the past is critical to managing drought risk and water crises in the future (*Zhang et al., 2012*; *Xu et al., 2015a*). This study aims to: (1) determine the spatio-temporal variations of drought in Xinjiang, northwestern China; (2) provide evidence of increasing drought severity caused by temperature rise; and (3) evaluate the response of NDVI and hydrology to drought.

## MATERIALS AND METHODS

The study region corresponds to Xinjiang, the largest province in China, which is located in the Eurasian hinterland. The province roughly spans the area between 73.66°E–96.38°E and 34.42°N–49.17°N and covers an area of $1.66 \times 10^6$ km$^2$. The province is primarily characterized by desert and grasslands, and contains three mountain ranges, the Tianshan,

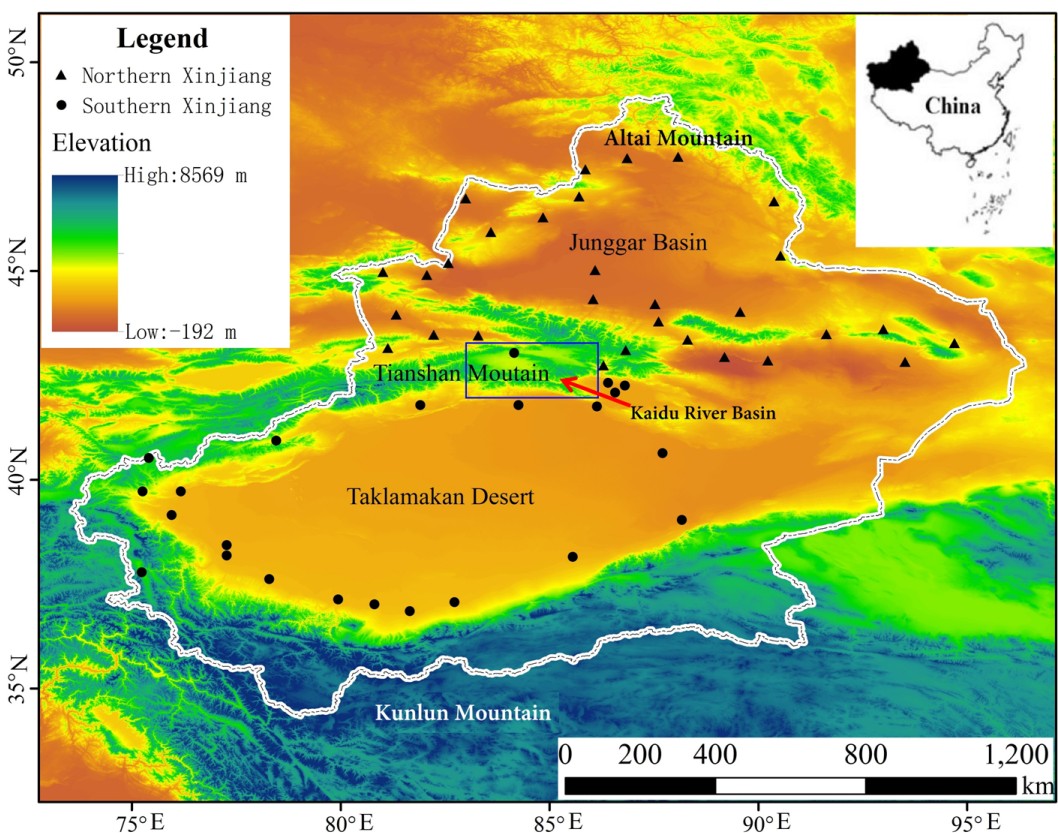

**Figure 1** **Study area and meteorological stations in Xinjiang (*Yao et al., 2018b*).** The blue box represents the Kaidu River basin.

Altai, and Kunlun Mountains, that are surrounded by vast desert basins (Fig. 1) (*Zhang et al., 2012*; *Wang et al., 2014*; *Chen et al., 2015*). Xinjiang is divided by the Tianshan Mountains into northern and southern Xinjiang, resulting in mountain-basin systems with different hydrological conditions (*Xu et al., 2004*). Xinjiang is far from any maritime influences and receives only about 158 mm of precipitation a year, making it one of the driest areas in the world (*Bai et al., 2014*). The precipitation distribution exhibits a clear summer pattern, reaching its maximum between May and August.

Continuously observed monthly temperature and precipitation readings from 76 stations in Xinjiang were provided by the China Meteorological Administration (CMA) for the period 1961–2015. To guarantee consistency, the monthly data were checked to ensure that they met the expected standards. The standard requires strict quality control processes including extreme inspection, time consistency check, and others before releasing these data. Pan evaporation data for northwestern China for 1961–2010 were obtained from *Li et al. (2013)*. Soil moisture data for the period 1980–2010 were provided by the National Meteorological Information Center (NMIC), and the depths of the five soil layers range from 0 to 10, 10 to 20, 20 to 30, 30 to 40, and 40 to 50 cm.

Streamflow and lake level data were obtained from the Dashankou station and from Bosten Lake, which are connected by the Kaidu River. This river descends from the southern slopes of the Tianshan Mountains (*Chen & Chen, 2014*) and runs through large irrigated areas into Lake Bosten. The Dashankou station is located at the foot of the Tianshan mountains.Data from both stations Observed streamflow and level data were provided by the Xinjiang Hydrology Bureau. The runoff coefficient, which is defined as the ratio between precipitation and streamflow over a given time period (*Chow, Maidment & Mays, 1998*), and the monthly runoff coefficient was calculated for the period 1961–2012.

Remote sensing is a very effective method for estimating important information about spatio-temporal changes in vegetation conditions (*Gómez-Mendoza et al., 2008*; *Coops et al., 2009*). The Normalized Difference Vegetation Index (NDVI) (*Rouse, 1974*) is widely used as a proxy indicator of vegetation change and can be derived from the Advanced Very High Resolution Radiometer (AVHRR) sensor on the National Oceanic and Atmospheric Administration (NOAA) Polar Operational Environmental Satellite (POES) series, among others (*Gómez-Mendoza et al., 2008*; *Eastman et al., 2013*). The newest version of the Global Inventory Modelling and Mapping Studies-NDVI third-generation (GIMMS-NDVI3g) datasets has been recently released with a spatial resolution of 8 km and a 15-day temporal frequency (*Zhu et al., 2013*). The GIMMS-NDVI3g data for Xinjiang were extracted for the period 1982 to 2013 and used in this study.

SPI and SPEI were used to quantify drought, as their multi-scalar character helps to determine drought variability. SPEI is calculated based on the climatic balance between monthly precipitation and atmospheric evaporative demand (monthly potential evapotranspiration, ET), whereas SPI includes only input monthly precipitation data. We employed the *Thornthwaite (1948)* model approach to calculate the ET. *Thornthwaite (1948)* correlated monthly mean temperature with PET, as determined from the water balance. *Willmott, Rowe & Mintz (1985)* modified Thornthwaite's original approach slightly by introducing parameterization for a limited range of average air temperature $T$ (Units: °C):

$$\text{PET} = \begin{cases} 0 & T < 0 \\ 16\left(\dfrac{N}{12}\right)\left(\dfrac{NDM}{30}\right)\left(\dfrac{10T}{I}\right)^{m} & 0 \leq T < 26.5 \\ -415.85 + 32.24T - 0.43T^2 & T > 26.5 \end{cases} \tag{1}$$

where $N$ is maximum number of sun hours, $NDM$ is number of days in the month, and $I$ is heat index, which is calculated as the sum of 12 monthly index values:

$$I = \sum_{i=1}^{12}\left(\frac{T}{5}\right)^{1.514} \quad T > 0; \tag{2}$$

and $m$ is a coefficient that depends on $I$:

$$m = 6.75 \times 10^{-7}I^3 - 7.71 \times 10^{-5}I^2 + 1.79 \times 10^{-2}I + 0.492. \tag{3}$$

The SPEI is quantified through the following steps: (a) calculating the PET; (b) determining the deficit or surplus accumulation of a climate-water balance (P-PET)

**Table 1 Drought classifications of SPI and SPEI.**

| Category | Index value |
|---|---|
| Extremely wet | Value $\geq 2$ |
| Moderately wet | $1.5 \leq$ value $< 1.99$ |
| Slightly wet | $1 \leq$ value $< 1.49$ |
| Near normal | $-0.99 <$ value $< 0.99$ |
| Mild drought | $-1.49 <$ value $\leq -1$ |
| Moderate drought | $-1.99 <$ value $\leq -1.5$ |
| Extreme drought | Value $\leq -2$ |

at different time scales; and (c) normalizing the water balance into probability distribution to obtain the SPEI series. In our study, Gamma distribution was chosen to calculate SPI for different time scales, and the SPEI index was calculated using the three-parameter log–logistic probability distribution.

SPI and SPEI were calculated for each month of the year, and time scales of 1, 3, 6, and 12 months were selected for analysis. The SPI and SPEI drought categories are listed in Table 1. Drought duration was calculated at each station for each year, and a threshold value of -1 was used to determine a drought condition. The duration of a drought was interpreted as the number of dry months. Regional drought durations were estimated for each year.

The nonparametric Mann-Kendall method (M-K) recommended by the WMO (*Mann, 1945*; *Kendall, 1975*; *Sneyers, 1990*) was used to investigate the temporal evolution of climate and drought indices from 1961 to 2015. This method has been used to investigate the significance of trends in various hydro-meteorological factors, including precipitation, runoff, evaporation and drought indices (*Zhang et al., 2012*; *Tao et al., 2014*; *Benitez & Domecq, 2014*; *Potop et al., 2014*; *Vicente-Serrano et al., 2014*; *Wang et al., 2015*; *Li, Chen & Yuan, 2015*). The Pearson correlation coefficient was used to investigate the relationship between drought indices and climate series.

## RESULTS

### Temporal variations in temperature and precipitation

From 1961 to 2015, annual mean temperature in Xinjiang, China, experienced a significant increasing trend, with a rate of increase of 0.31 °C/10a. This is considerably higher than the rates of increase observed for China as a whole, or globally (0.25 °C/10a and 0.12 °C/10a, respectively) (*IPCC, 2013*). The warming trend started to intensify during the late 1980s, then increased sharply in 1997 (Fig. 2A). During the recent 17-year period (1997–2015), temperature has been in a highly volatile state, making this the warmest period in the last half-century. However, although temperatures remained elevated during this period, there was no clear increase. As a result, the period from 1997–2015 is now referred to as a "hiatus" (*Easterbrook, 2008*; *Easterling & Wehner, 2009*; *Kaufmann et al., 2011*). Spatially, the main areas of accelerated warming appear to be in northern Xinjiang. Over the past half-century, precipitation has exhibited a slight increasing trend and greater decadal fluctuations.

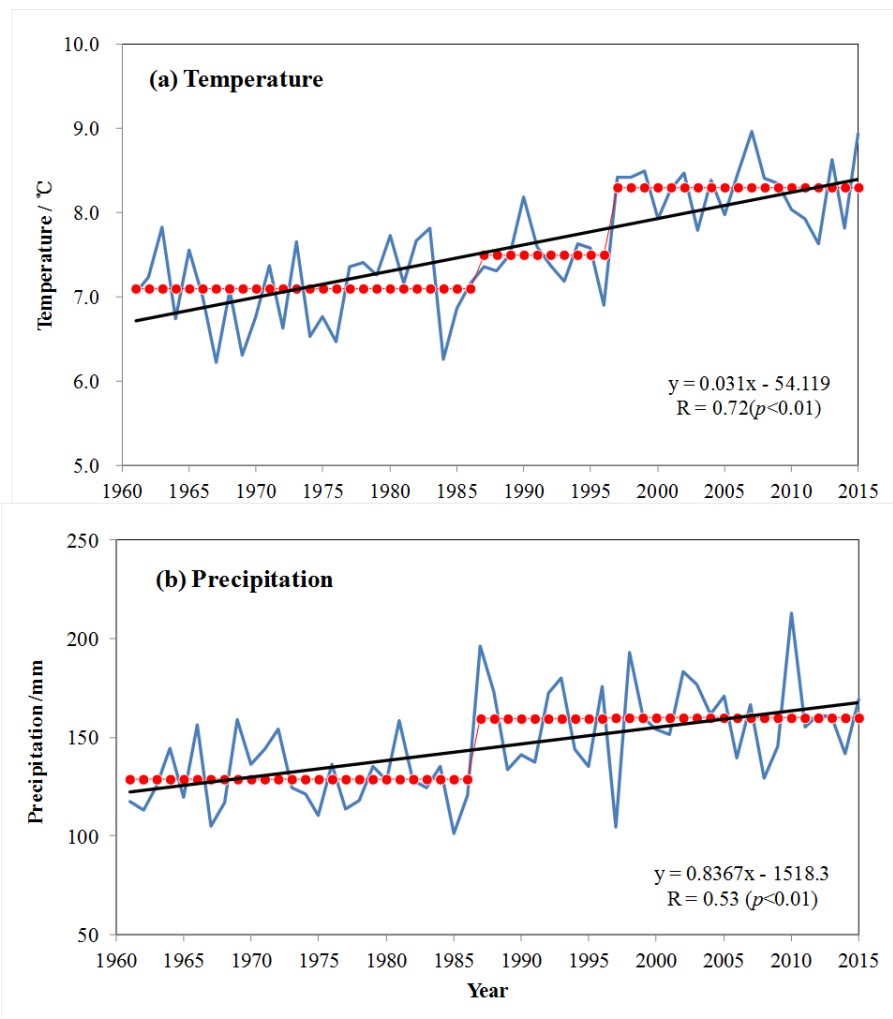

**Figure 2** Temporal changes of (A) annual mean temperature (B) and annual precipitation in Xinjiang during 1961–2015.

Annual precipitation remained relatively stable from the 1960s to the mid-1980s, then began to increase sharply in 1987 (Fig. 2B). The most humid decade was the 2000s, but without the continuously increasing trend of the 1990s. Spatially, the area with the most significant increase in precipitation was mainly in northern Xinjiang. Some researchers have suggested that these data show a warmer and wetter trend for Xinjiang (*Shi et al., 2003*; *Fang et al., 2013*; *Chen et al., 2015*).

## Relationship between the SPI and SPEI drought indices

Monthly SPI and SPEI were estimated at four time scales (1, 3, 6, and 12 months) for all selected stations from 1961 to 2015. The average SPI and SPEI were then calculated to characterize drought episodes in Xinjiang (Fig. 3). According to historical records, the evolution of SPI and SPEI is highly consistent between stations and time scales (*Cao, Nan & Cheng, 2015*). Figure 4 displays the difference between the two drought indices at four

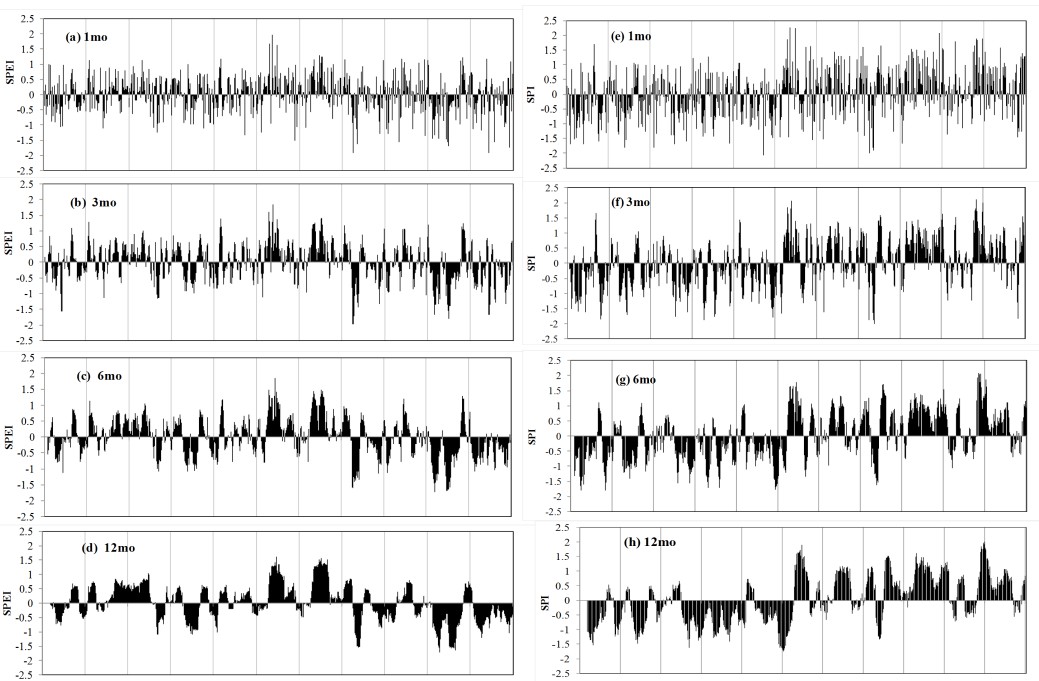

**Figure 3** Evolution of the 1-, 3-, 6-, and 12- month SPEI (A–D) and SPI (E–H) for Xinjiang, China.

lags. It illustrates that SPEI became gradually lower than the corresponding SPI and that the difference has increased in recent decades. Moreover, these differences were more evident over 6- and 12-month lags than over 1- and 3-month lags. In late 1996, the difference became as large as −1.5.

On a 6-month time scale, SPI and SPEI showed large differences before the 1990s (Fig. 4C). More wet events and fewer dry events were identified based on SPEI. Nevertheless, based on SPEI, most of the 1990s and the first decades and a half of the 21st century qualify as some of the most drought-intense years on record. On a 12-month time scale, both indices identified the major drought periods from 1961 to 1966 and from 1976 to 1986. However, only the SPEI was able to identify drought events between 1996 and 2015 (Fig. 4D). In general, SPEI confirmed that the most frequent droughts occurred in the 1960s, 1970s, late 1990s, and late 2000s, whereas SPI identified drought conditions during the 1960s, 1970s, and early 1980s, as well as humid periods.

The increase in temperature enhanced ET, causing excessive water consumption and thus lowering the value of SPEI. In arid regions, water consumption occurs mainly through actual evapotranspiration (AE/mm), as there is ample energy (radiation) available for evaporation. For the 21st century, the number of drought episodes identified by SPEI was greater than the number identified by SPI due to the enhanced evaporation amount.

Figure 5 shows the Pearson's *r* coefficients between SPI and SPEI estimated at 1-, 3-, 6-, and 12-month lags in the series for all of Xinjiang, northern Xinjiang, and southern Xinjiang from 1961 to 2015. Most observations showed a high correlation between SPI and SPEI. The correlations among the drought indices was notably stronger over shorter time

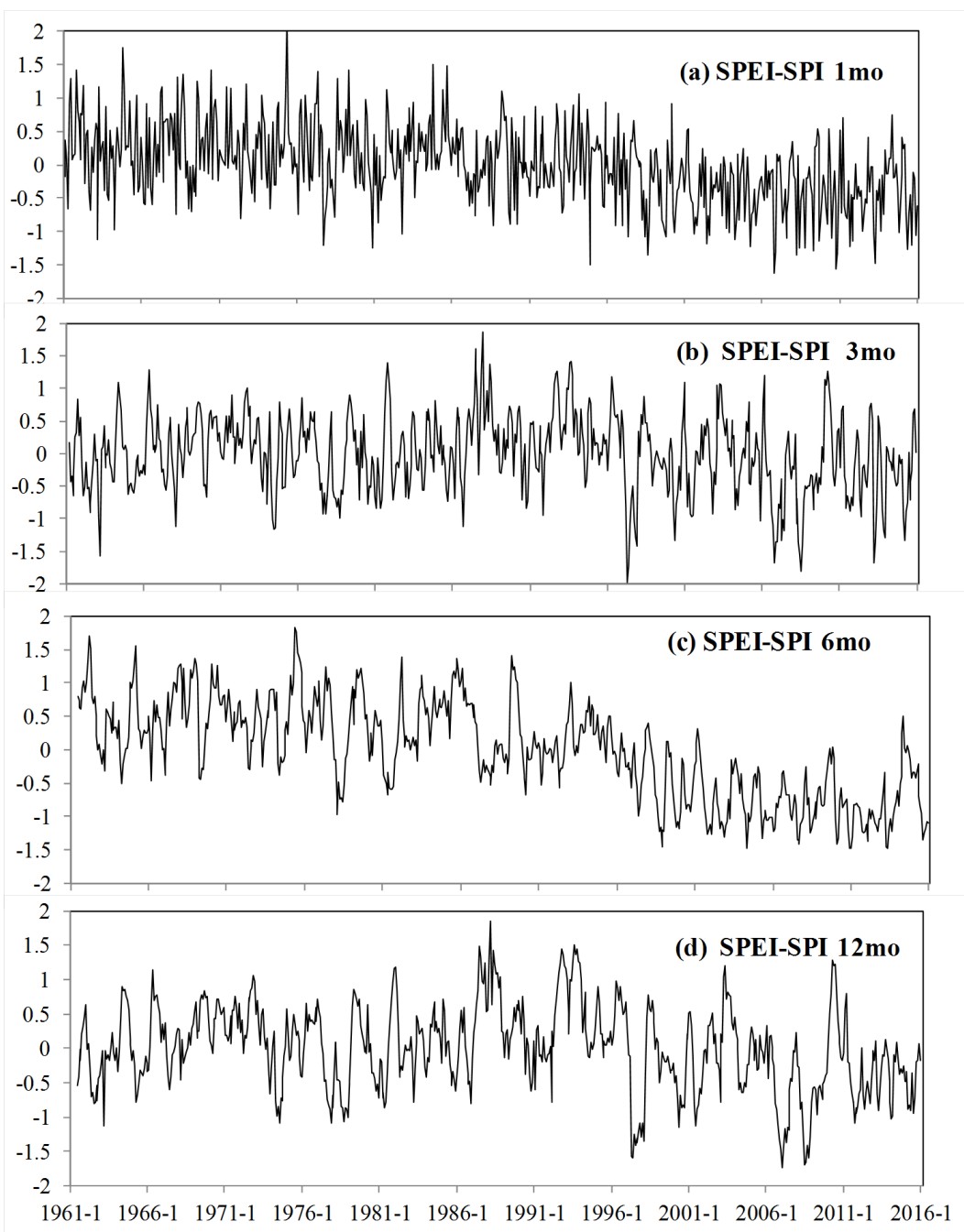

**Figure 4** The difference between SPEI and SPI at 1- (A), 3- (B), 6- (C) and 12- (D) month lags, the ordinate denotes the value using SPEI minus SPI.

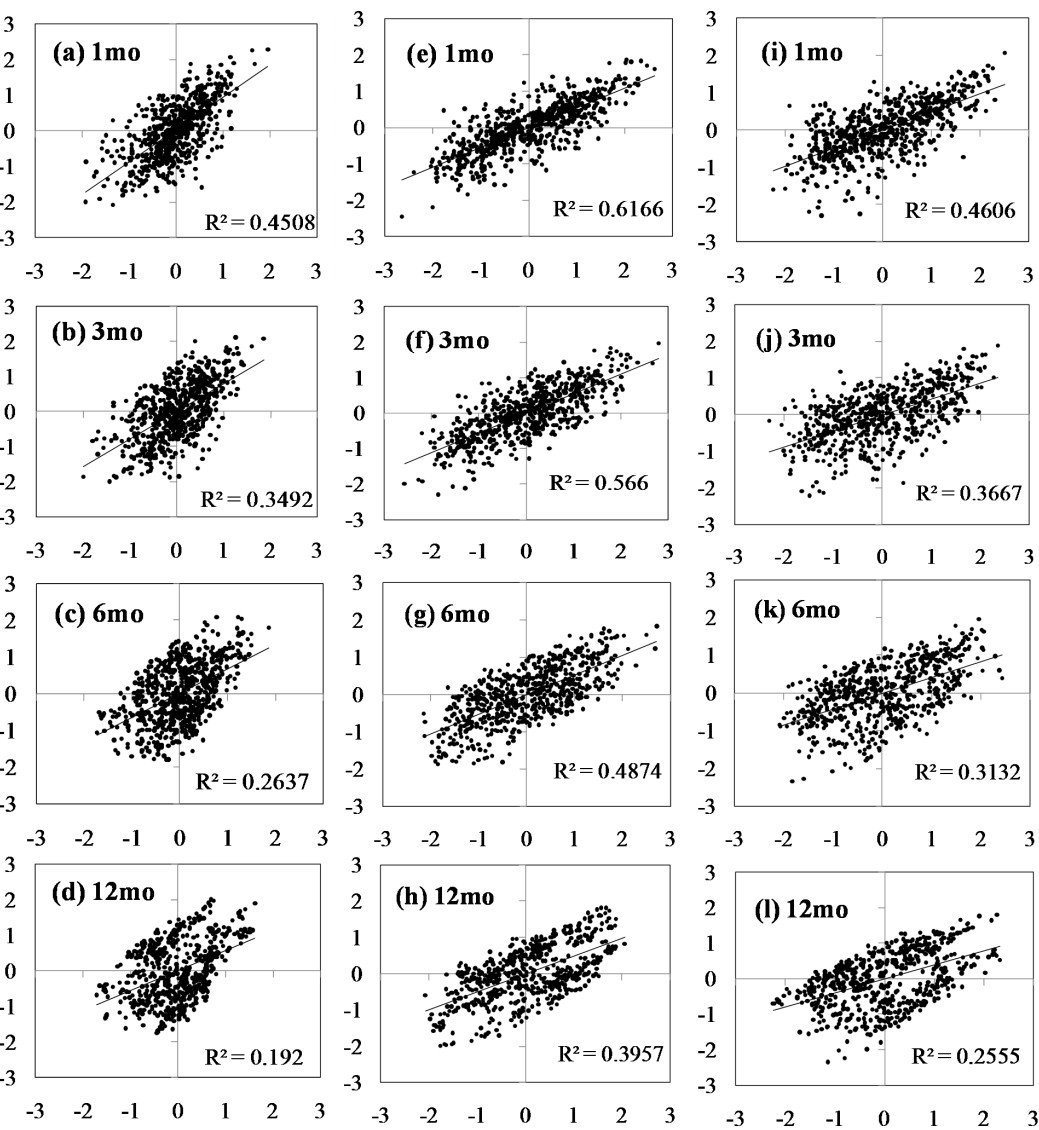

**Figure 5** Pearson's r correlations between the time series of the two drought indices of the 1-, 3-, 6-, and 12- month SPI (X axis) and SPEI (Y axis) for the whole Xinjiang (A–D), Northern Xinjiang (E–H) and Southern Xinjiang (I–L) during the 1961–2015.

scales than over longer time scales, varying between 0.67 for a 1-month lag and 0.44 for a 12-month lag for the whole region. Spatially, the correlation between 12-month SPI and SPEI was very high in Xinjiang, with 86% of observations reaching the 99% significance level. Correlations between SPEI and SPI were notably stronger in northern Xinjiang than in southern Xinjiang at all four time scales (Fig. 5).

## Evidence of increasing drought caused by temperature rise

The 12-month SPI and SPEI series showed coherent variation in Xinjiang before 1997 (Fig. 6). A major drought in Xinjiang was observed from 1975 to 1986, after which both

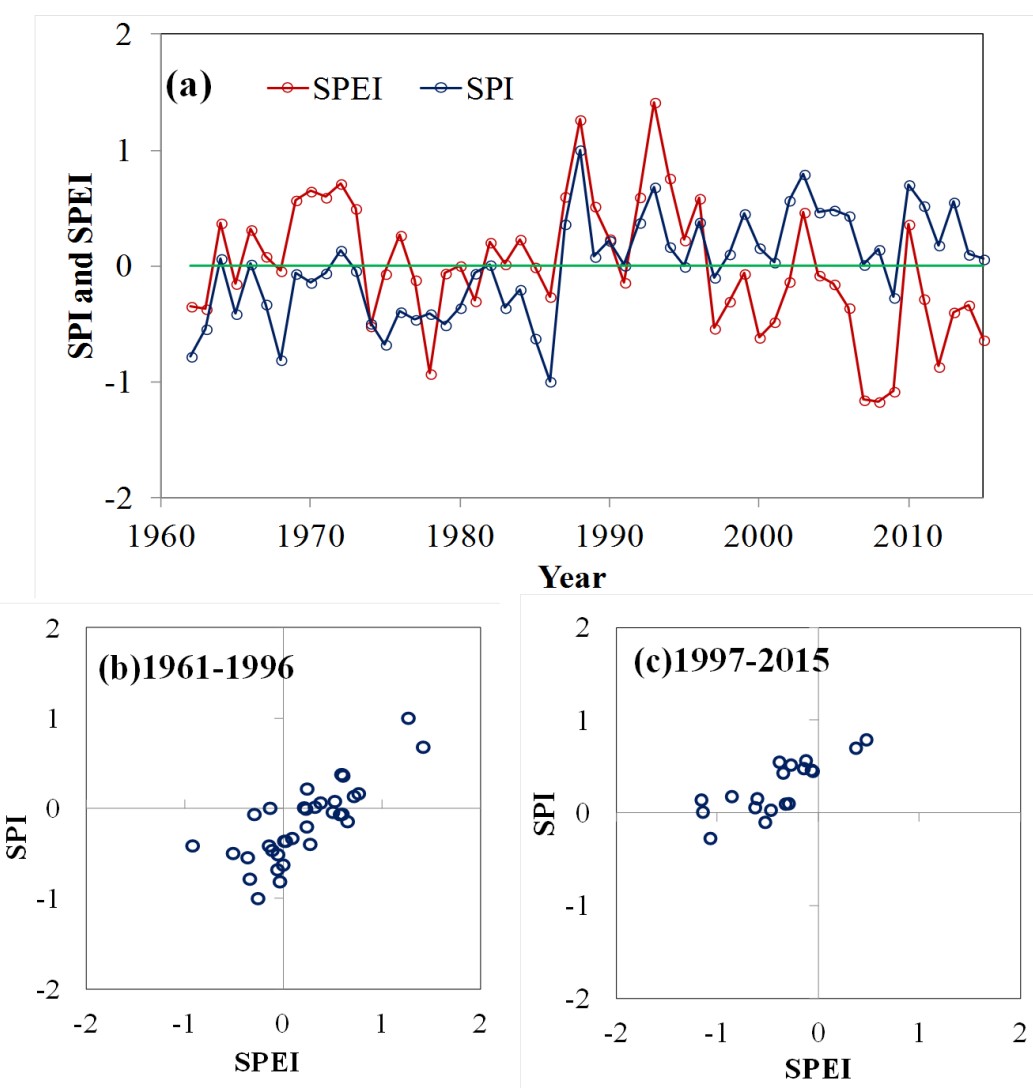

**Figure 6** Evolution of the regional SPI (blue line) and the SPEI (red line) for the Xinjiang from 1961 to 2015 (A); the scatter diagram of the SPI and SPEI at the both episodes: (B) 1961–1996 and (C) 1997–2015.

indices identified smaller drought events between 1987 and 1996, with a trend towards more positive values. Nevertheless, SPEI also identified increased drought severity relative to SPI between 1997 and 2015 (Fig. 6A and 6C). The increased drought severity is due mainly to the temperature rise in Xinjiang, which is better captured by SPEI than by SPI.

Figure 7 shows the spatial patterns of the 12-monthly SPI and 12-monthly SPEI trends for the 1961–1996 and 1997–2015 at each station. From 1961 to 1996, SPI evolved in a positive direction (meaning wetter conditions) over Xinjiang; increased drought was recorded at only a few stations (Fig. 7A). For 12-monthly SPEI, about 72.5% of stations showed increasing trends (Fig. 7C). SPI showed increasing drought conditions from 1997 to 2015, during which time about 47.1% of stations in northwestern and southwestern

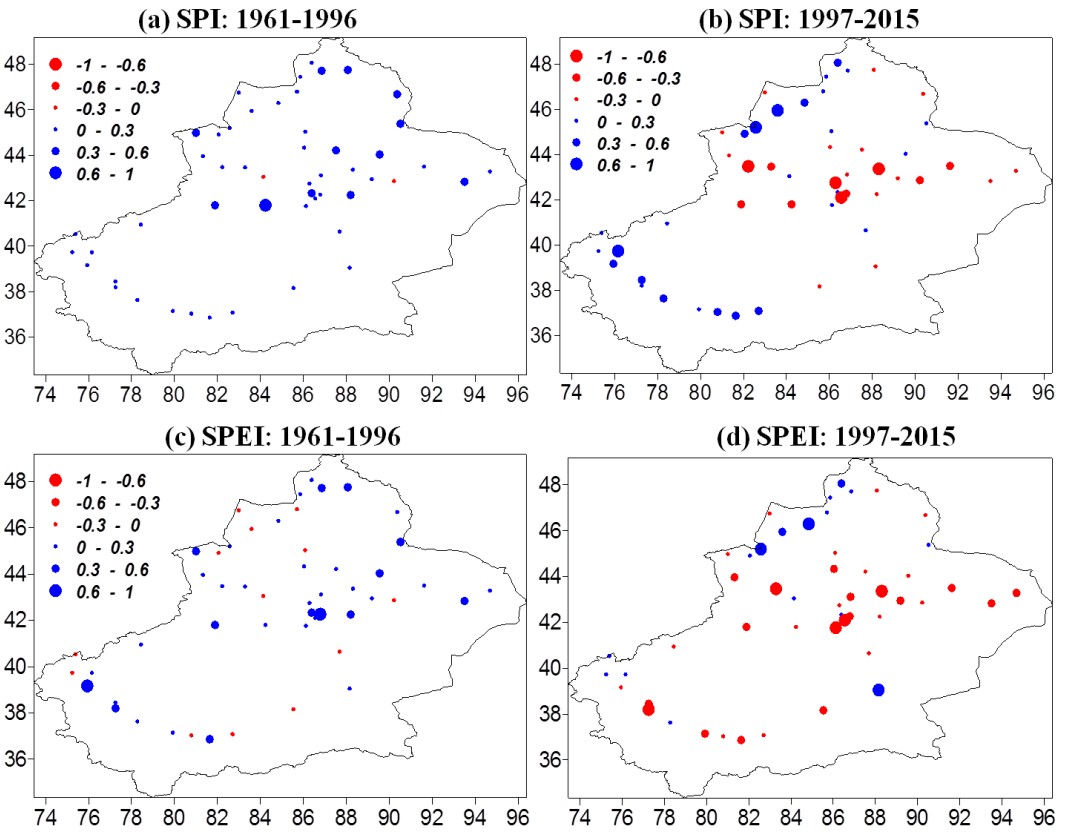

**Figure 7** The spatial distribution of the temporal trends per decade in 12- monthly SPI and the 12-monthly SPEI for the period 1961–1996 (A–B) and 1997–2015 (C–D) in Xinjiang.

Xinjiang had larger trend magnitudes (Fig. 7B). SPEI indicated a more obvious decrease than SPI at 70.5% of stations (Fig. 7D). Furthermore, northwestern Xinjiang and Kunlun Mountain display obvious wetting trends during this period. Hence, both indices indicated increasing drought severity in Xinjian over the last two decades, but SPEI identified a larger drought region than SPI, which neglects the effect of ET.

The main driving force underlying evaporation changes in arid China is the aerodynamic component, specifically wind speed, while the radiative component has a minor impact (*Li et al., 2013*). Observed vapor pressure deficit (VPD) was also a primary contributor to ET changes after the early 1990s (*Li et al., 2013*). The VPD exhibited an obvious increasing trend, with a rate of 3.9 mm/year from 1994 to 2010 (*Li et al., 2013*). This trend reduced moisture supply and increased the atmospheric water-holding capacity as a result of the increase in temperature. The strengthened evaporative demand was evidenced by trends in potential evaporation in Xinjiang, which were observed directly by pan evaporation (Fig. 8). Annual pan evaporation data exhibit a strong decreasing trend with a rate of −6.0 mm/year from 1975 to 1993. Since 1994, however, this trend has reversed upward at a rate of 10.7 mm/year.

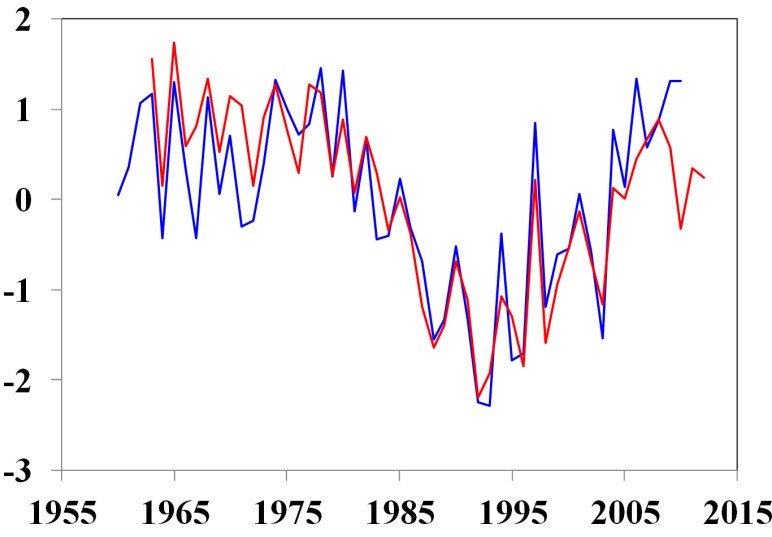

**Figure 8** **Evolution of the standardized Thornthwaite ET (red line) and pan evaporation (blue line) in Xinjiang.** The Pearson's r correlation for the common period (1961–2010) was 0.87 ($p < 0.01$).

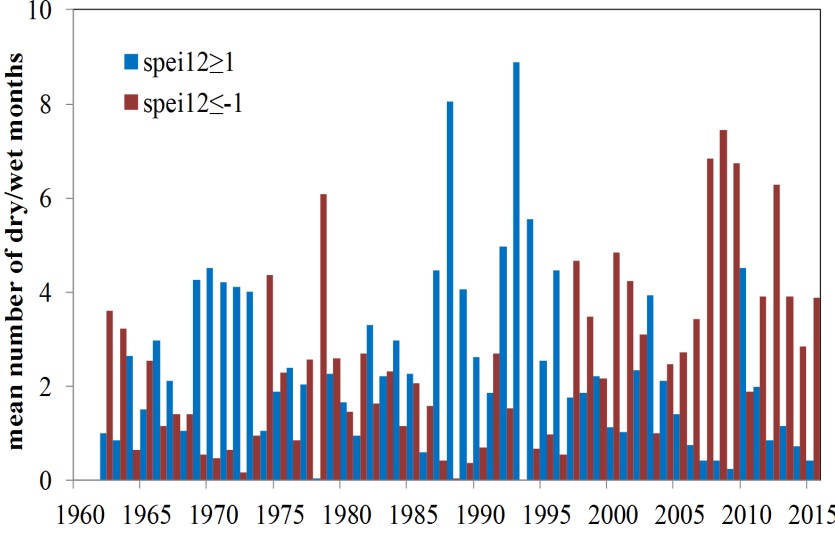

**Figure 9** **The mean number of dry (SPEI $\leq -1$) and wet months (SPEI $\geq 1$) each year by the 12-month time scale from 1962 to 2015.**

## Spatio-temporal extent of drought

Figure 9 illustrates the average number of dry or wet months in each year from 1961 to 2015. Relatively long series of dry months appeared in the early 1960s, late 1970s, and early 1980s, and very long series of dry months occurred in the late 1990s until the 2010s. In contrast, a several series of wet months occurred in the early 1970s, and a very long series of wet months occurred from the late 1980s until the middle of the 1990s. The temporal variation in the number of dry months showed a significantly increasing trend

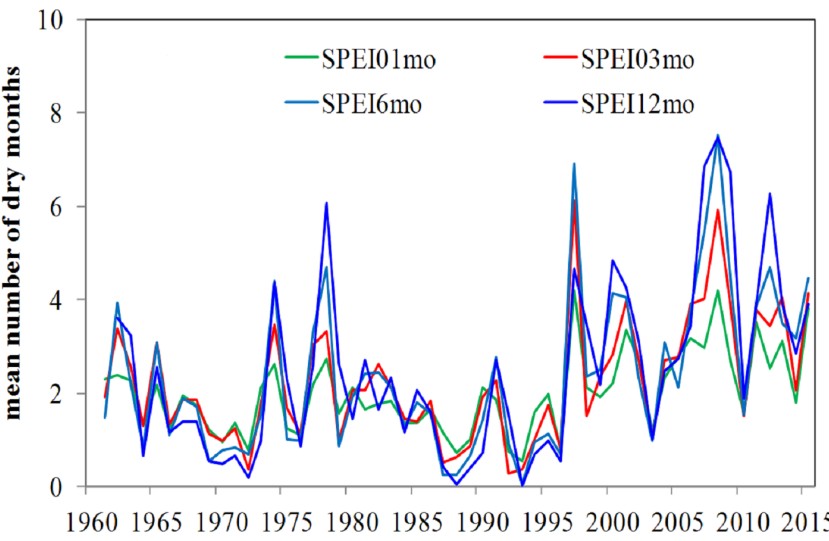

**Figure 10  The mean number of dry months (SPEI $\leq -1$) each year from 1961 to 2015.**

(0.54 mo/10a, $p < 0.01$). Before 1997, there were fewer than two dry months per year but the number of dry months has increased since then. The average number of dry months in recent years was greater than four, and the maximum number of dry months in a year was greater than seven (from 2007 to 2009).

Figure 10 shows the mean number of dry months per year at different SPEI time scales from 1961 to 2015. The results from the two indices are generally comparable before the late 1990s. The number of droughts identified at 1-, 3-, and 6-month time scales was slightly less than at the 12-month time scale. However, a sharp transition occurred by the late 1990s. The average number of dry months for SPEI-12 was four, which was greater than the numbers for shorter SPEI time scales (3.5 months for SPEI at 3 and 6 months, and 2.7 months for SPEI at 1 month). However, because the high frequency noise is attenuated over longer periods, it is expected that the 12-month SPEI will have a higher number of dry months. Especially between 2007 and 2009, the number of dry months in SPEI-12 greatly exceeds the shorter accumulation periods (7.0 months for SPEI-12, 5.2 months for SPEI-3 and SPEI-6, and 3.3 months for SPEI-1). Periods during which temperature increases (and precipitation decreases or remains constant) will probably increase this difference.

Next, the percentage of stations with dry months (SPEI $\leq -1$) was considered to investigate the spatial variation of drought severity. Figure 11 shows the variation in the percentage of stations with more than 3 and 6 dry months at a 12-month time scale during the period 1961–2010. The other cases are not listed here. A significant increasing trend in the percentage of stations with more than three dry months was found in Xinjiang, with a rate of increase of 5.80%/decade (Fig. 11). The highest number of droughts occurred after 1997, when about 54.93% of stations experienced drought. Widespread and severe droughts (those for which the percentage of stations recording drought exceeds 60%) were recorded in 1962, 1974, 1978, 1997–1998, 2000–2001, 2007–2009, and 2012, with 62%,

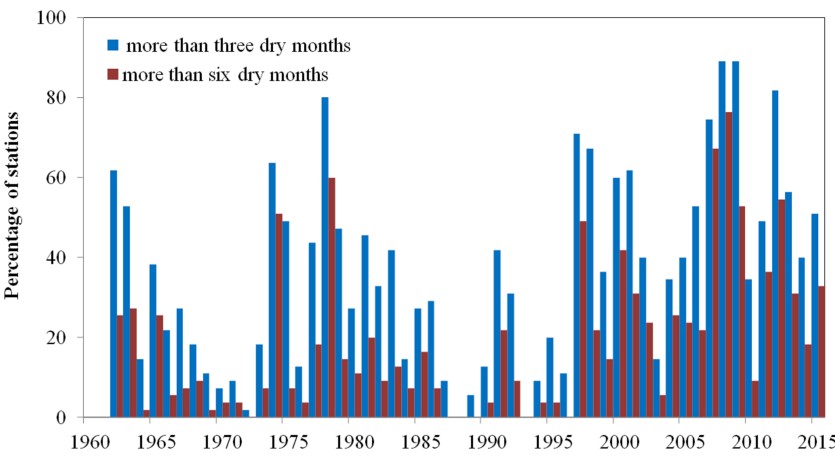

**Figure 11** Percentage of stations with more than 3 dry months and more than 6 dry months each year.

64%, 80%, 71%, 67%, 60%, 61%, 75%, 89%, 89%, and 82% of stations recording drought, respectively.

This analysis shows that drought conditions have been aggravated over the last two decades in Xinjiang. The percentage of stations with more than six dry months also showed a significant increasing trend, with a rate of 4.91% /decade ($p < 0.05$) (Fig. 11). During the period 2007–2008 in particular, more than 71% of stations sustained drought conditions for more than six months.

Figure 12 shows the severe drought events mentioned above. The 1974 drought mostly affected northern Xinjiang, particularly the northern slopes of the Tianshan Mountains, Urumqi, the Changji region, and the Altay region (*Shi, 2006*). The 2008 drought was one of the most serious droughts in recent times with regard to drought duration, area affected, and agricultural, livestock, and economic losses. The drought, which lasted from spring to autumn, was mainly located in southern Xinjiang, the Tianshan Mountain region, the Yili Valley, Tacheng, Altay, and the Hami region. It was reported that the drought caused the loss of 1.22 million hectares of crops, 28 million hectares of grasslands, and one billion yuan in direct economic losses (*Cao, Nan & Cheng, 2015*; *CMA, 2010*). The 2009 drought was persistent and affected southern Xinjiang, while a slightly less severe drought occurred in the eastern and northwestern parts of Xinjiang. The drought caused the Tarim River, China's longest inland river, to run dry over more than 1100 kilometers (*CMA, 2010*). In the spring and summer of 2012, northern Xinjiang and the Tianshan Mountains experienced severe droughts, particularly in the Yili, Tacheng, and Altay regions. The droughts reduced the yield of industrial tomatoes (in one of the world's three largest industrial tomato-growing areas) to less than half that of the year before (http://finance.sina.com.cn/roll/20121016/110213381605.shtml).

## Response of vegetation NDVI and hydrology to drought

Increased meteorological drought severity should be evident in increased impacts on drought-sensitive systems, especially natural vegetation. The NDVI reflects ecosystem

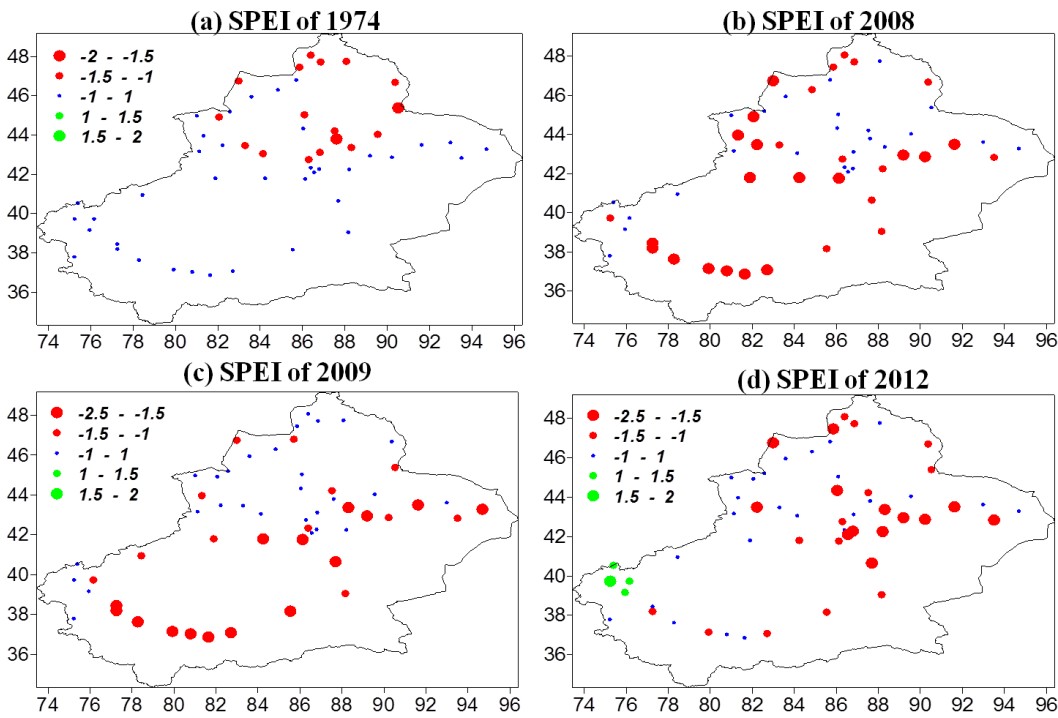

**Figure 12** Spatial variations of severe drought events: (A) 1974, (B) 2008, (C) 2009, and (D) 2012.

processes and is a critical indicator for monitoring climate dryness and wetness. *Yao et al. (2018a)* analyzed the response of vegetation NDVI to climate change over Xinjiang from 1982 to 2013 and found that the NDVI increased from 1982 to 1997, then decreased significantly after 1997.

The results of this study confirm a strong negative correlation ($r = -0.52$; $p < 0.01$) between NDVI and an increase in PET, but the correlation between NDVI and precipitation does not show a clear relationship. Nevertheless, the relationship between the NDVI and SPEI ($r = 0.37$; $p < 0.01$) shows a stronger correlation than that between the NDVI and SPI ($r = 0.22$; $p < 0.05$), implying that the decrease in vegetation NDVI is mainly controlled by PET. PET influences vegetation NDVI by affecting soil moisture and plant transpiration, which eventually leads to soil moisture loss (*Li, Chen & Yuan, 2015*; *Li et al., 2015a*; *Li et al., 2015b*; *Li et al., 2015c*). Soil moisture in Xinjiang has recently shown a marked decreasing trend, which is especially obvious in shallow soil layers (Fig. 13 and Table 2). Climate warming increases soil moisture loss and cause the death of shallow-rooted plants (*Li, Chen & Yuan, 2015*). Consequently, NDVI decrease and soil moisture loss have almost the same effect as climate dryness.

Drought trends and their relation to water resources are directly related to hydrological drought in China (*Shen & He, 1996*; *Zhai et al., 2010*). Drought at the 12-month time scale is relevant for hydrological impacts (*Benitez & Domecq, 2014*). Runoff in some of the larger rivers in arid China has shown a widespread "sharp" increasing trend from 1961 to 2010 (*Chen et al., 2015*), and a "sharp" increase in the early 1990s, especially in the

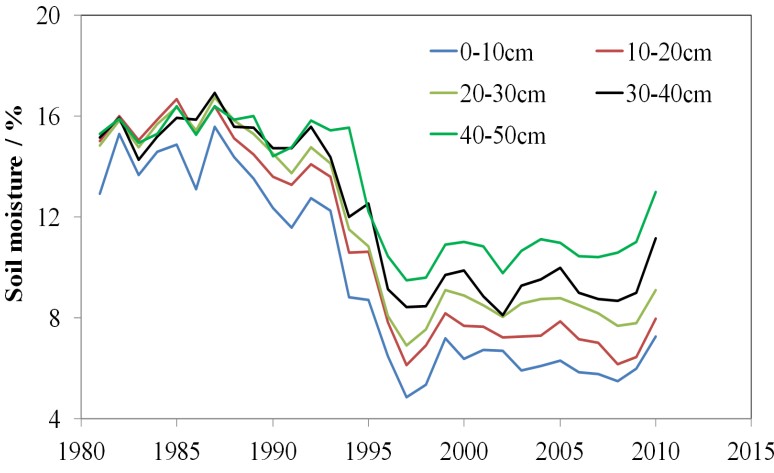

**Figure 13** Annual variation of soil moisture at each depth (0–50 cm) in Xinjiang, China.

**Table 2** Mean value and tendency of soil moisture at each depth (0–50 cm) in Xinjiang.

| Depth (cm) | 0~50 | 0~10 | 10~20 | 20~30 | 30~40 | 40~50 |
|---|---|---|---|---|---|---|
| Mean Value (%) | 11.3 | 9.5 | 10.7 | 11.5 | 12.0 | 12.9 |
| Trend (%/10a) | −3.8 | −3.6 | −4.0 | −3.5 | −3.0 | −2.3 |

Tarim River basin (*Chen et al., 2015*). In this study, long series of runoff data for typical rivers in Xinjiang were analyzed, including for the Kaidu River (one of the four main source streams of the Tarim River), which has not been influenced by human interference and regulation. This analysis discovered a significant upward trend (11.2 mm/ decade, $p < 0.01$) in measured runoff in the Kaidu River from 1960 to 2010 (Fig. 14B), with a 26.5% increase after the early 1990s. Figures 14A and 14B displays the evolution of runoff and precipitation in the Kaidu River basins, and there have a strong correlation ($r = 0.70$; $p < 0.01$). Nevertheless, the runoff coefficient shows a significant increase since 1990 (Figure 14C), implying that runoff has increased more than precipitation. Figure 15 presents the positive relationship between PET and the runoff coefficient of 68 rivers in arid China. This coincides with a marked increase in mountain temperature and drought indices (Fig. 14D), suggesting that increased meltwater from glaciers and snowpack may be contributing to incremental runoff. Furthermore, it is noteworthy that the rate of increase in precipitation has diminished since the beginning of the 21st Century (Fig. 14A). Hence, the decrease in drought indices is consistent with decreased P (Fig. 14D). Lake Bosten, China's largest inland freshwater lake, is the outlet lake of the Kaidu River. The decrease in observed lake level has become more acute since the start of the 21st century, coinciding with the evolution of increased drought (Fig. 14E). In addition, these developments are associated with large lake areas that favor direct evaporation under increased PET.

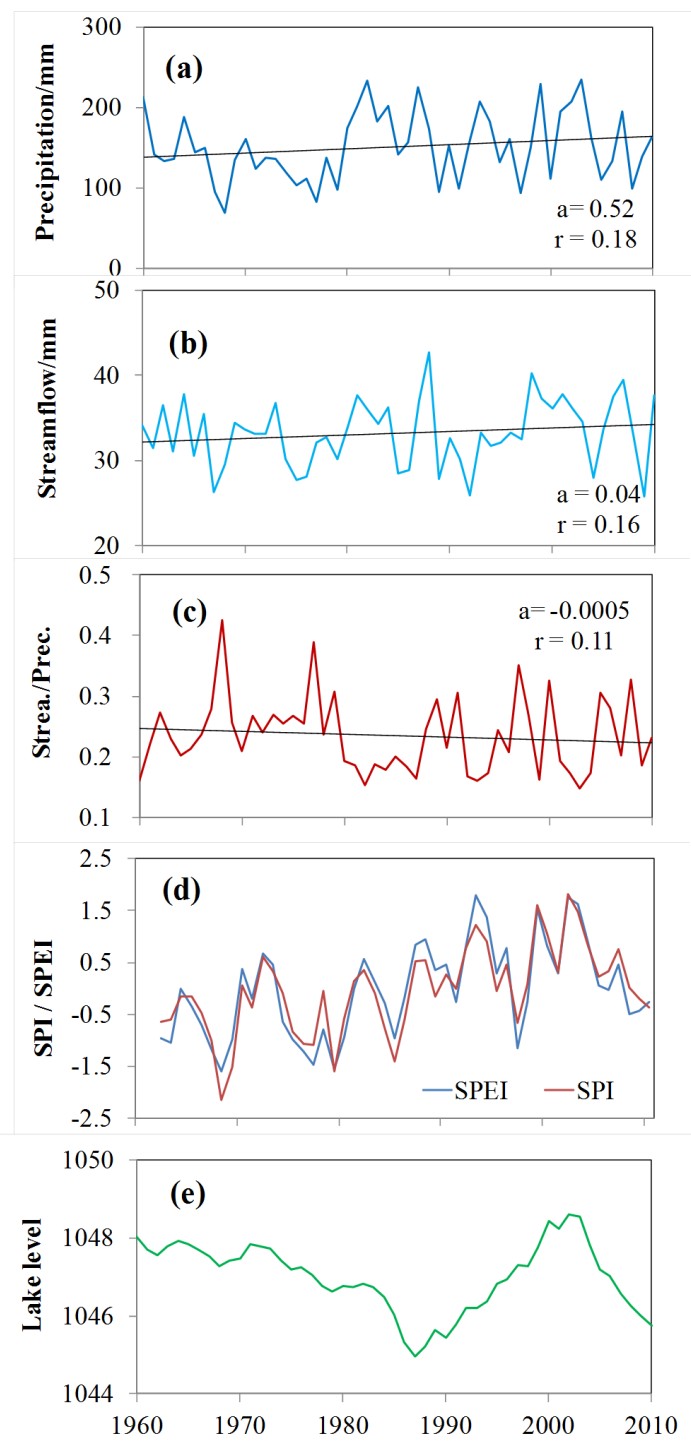

**Figure 14 Evolution of parameters in the Kaidu River basins of the southern Xinjiang, China.** Evolution of parameters in the Kaidu River basins of the southern Xinjiang, China, including: (A) total annual precipitation; (B) total annual streamflow at the Dashankou hydrological station; (C) runoff coefficient (ratio between precipitation and streamflow); (D) SPI and SPEI at the Bayinbuluke climate station; and (E) water level at the Bosten Lake.

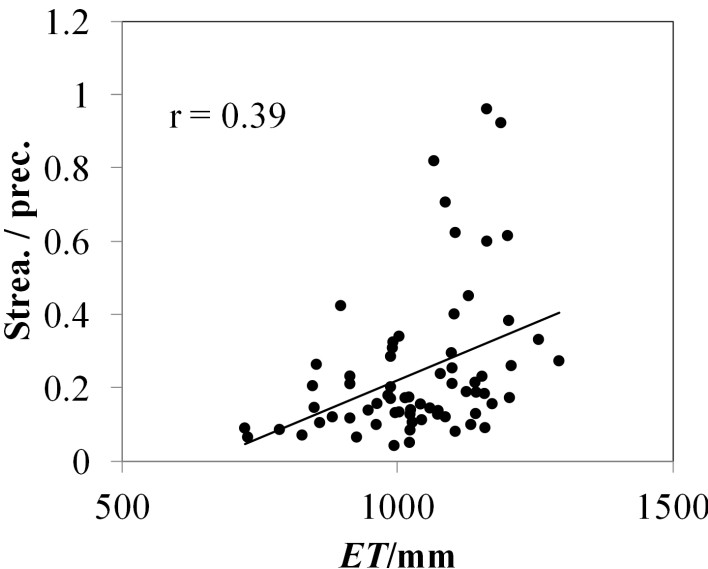

**Figure 15** The scatterplot between annual ET and the annual streamflow/precipitation ratio of the 68 inland river catchments in northwestern China.

## DISCUSSION

Climate change in Xinjiang is complex and responds sensitively to global warming (*Chen, Yang & Luo, 2012*; *Xu et al., 2013*; *Chen & Sun, 2015*). *Shi, Shen & Hu (2002)* found a climatic transition from warm-dry to warm-wet in Xinjiang in the early 21st century based on observed data. Most evidence for hydroclimatic and environmental change in Xinjiang since then has strongly supported this observation (*Shi et al., 2003*; *Shi et al., 2007*; *Zhao, Yu & Chen, 2009*; *Xu et al., 2010*; *Li et al., 2011*; *Li, Chen & Shi, 2012*; *Li et al., 2013*; *Yao et al., 2017*). *Fang et al. (2013)* further confirmed this climatic transition in Xinjiang based on Coupled Model Inter-comparison Project-Phase 5 (CMIP5) model simulation results. We provide a latest observed data of climate change under global warming. The data indicate that warming started to accelerate during the late 1980s and underwent a sharp increase in 1997, after which temperatures remained highly stable (the warming hiatus). Meanwhile, precipitation exhibited a sharp increase in 1987, but has remained in a relatively stable state since then. As mentioned earlier, temperature and precipitation have exhibited a dramatic change under global warming.

*Li & Sun (2016)* and *Li et al. (2016)* demonstrated that drought relief resulted from the local climate becoming warmer and wetter in Xinjiang, as indicated by the precipitation-based drought indices. *Zhang et al. (2012)* also confirmed that SPI-based evaluations of drought severity in Xinjiang have lessened in recent years. In contrast, the results of this study indicate that SPEI-based drought severity has been aggravated in the last two decades as a result of temperature rise. The Xinjiang region has been experiencing increased drought severity since 1997 as a consequence of a significant increase in temperature (1.1 °C for two decades) coupled with an insignificant increase in precipitation (3.5 mm/10a). Drought

conditions have been exacerbated by higher evaporative demand, which increased by 10.7 mm annually from 1994 to 2010.

The major difference between the precipitation-based indices and SPEI is the meteorological factors used to calculate the indices, i.e., P and P-PET respectively. SPI considers only climatic water supply, whereas SPEI considers both climatic water supply and demand (*Xu et al., 2015b*). Precipitation is a critical meteorological driver of drought, but rising temperatures are playing an increasingly strong role in influencing drought severity (*Xu et al., 2015b*; *Vicente-Serrano et al., 2014*). SPEI is therefore a better indicator than SPI under climatic warming because it is sensitive to temperature.

Climate change effects on droughts have been found to exhibit similar characteristics at specific sites. *Vicente-Serrano et al. (2014)* suggested that drought severity has increased in southern Europe as a consequence of temperature rise. *Sun & Ma (2015)* also reported increasing drought severity over the Loess Plateau in China because of the combined effects of a significant increase in average temperature and an insignificant decrease in precipitation. The results described here are consistent with these results. This pattern of drought severity caused by temperature rise is probably applicable to other arid and semiarid areas of the world.

## CONCLUSIONS

In this research, two multi-scalar drought indices were used: the Standardized Precipitation Index (SPI) and the Standardized Precipitation-Evapotranspiration Index (SPEI). The evolution of meteorological drought was investigated in Xinjiang, northwestern China, for the period 1961–2015. This study provides evidence for increasing drought severity caused by temperature rise and discusses the effects on vegetation and water resources. A summary of the main results follows:

(1) Temperature and precipitation in Xinjiang exhibited a dramatic change between 1961 and 2015. Temperature experienced a sharp increase in 1997, after which it has remained highly volatile and exhibited a recent warming hiatus. Precipitation showed a sharp increase in 1987, but the increasing trend has diminished over the past two decades.

(2) The evolution of SPI and SPEI displayed a high level of agreement for most stations and time scales. SPI and SPEI showed increasing trends before 1997, after which the trends reversed. Between 1997 and 2015, SPI showed increasing drought conditions at 47.1% of the stations in northwestern and southwestern Xinjiang, especially at stations in the southwestern Tarim River Basin. SPEI showed increased drought severity across 70.5% of stations.

(3) SPEI-based drought severity was significantly aggravated relative to that based on SPI, which is independent of the effect of evaporative demand. Potential evaporation exhibited a strong increasing trend at a rate of 10.7 mm/year. Meteorological drought severity over the last two decades has been exacerbated by greater evaporative demand caused by a significant increase in temperature (1.1 °C) coupled with an insignificant

increase in precipitation (0.35 mm annually). Increased meteorological drought severity has had a direct effect on NDVI and river discharge.

### Funding
This work was supported by the National Natural Science Foundation of China (41605067); Basic Research Operating Expenses of the Central level Non-profit Research Institutes (IDM201506); National Natural Science Foundation of China (41375101); China Postdoctoral Science Foundation (2016M592915XB). The funders had no role in study design, data collection and analysis, decision to publish, or preparation of the manuscript.

### Grant Disclosures
The following grant information was disclosed by the authors:
National Natural Science Foundation of China: 41605067, 41375101.
Basic Research Operating Expenses of the Central level Non-profit Research Institutes: IDM201506.
China Postdoctoral Science Foundation: 2016M592915XB.

### Competing Interests
The authors declare there are no competing interests.

### Author Contributions
- Junqiang Yao conceived and designed the experiments, performed the experiments, analyzed the data, contributed reagents/materials/analysis tools, authored or reviewed drafts of the paper, approved the final draft.
- Yong Zhao conceived and designed the experiments, contributed reagents/materials/-analysis tools, authored or reviewed drafts of the paper, approved the final draft.
- Xiaojing Yu analyzed the data, prepared figures and/or tables, authored or reviewed drafts of the paper, approved the final draft.

### Data Availability
The raw data are provided as a Supplemental File.

### Supplemental Information
Supplemental information for this article can be found online at http://dx.doi.org/10.7717/peerj.4926#supplemental-information.

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
