# Peer review of "Spatial-temporal variation and impacts of drought in Xinjiang (Northwest China) during 1961–2015"

_PeerJ, doi:10.7717/peerj.4926_

## Round 0.1 · original submission · Major Revisions

· Academic Editor

Major Revisions

In view of the reviewers' comments, the current manuscript is worth further consideration. I would encourage the authors take the reviewers' comments seriously and revise the manuscript accordingly.

Reviewer 1 ·

Basic reporting

The paper is clear and unambiguous, professional English is used throughout.

The references are fine.

Some minor suggestions were made to a few of the figures in my review.

This paper is self contained.

Experimental design

Perhaps the only issue could be that the paper is from the Physical Sciences, but with implications to the Environment, and therefore is within the Scope of the Journal.

Research question is not that clearly stated though.

The research is rigorous.

Methods are described in detail.

Validity of the findings

All this is fine.

Additional comments

Review of “Evaluation of drought in Xinjiang, Norwest China: Spatial-temporal variation and uncertainty analysis”

By Yao et al.

PeerJ

General comments

This paper uses drought indexes to identify changes in a vast region of China. Trends and relationships between the indexes and also to precipitation and temperature are presented. I found the paper interesting and worth publishing with major corrections, that I will describe next.

The first comment that I have is that the title suggests some type of uncertainty analysis and I was expecting something else, not just the simple statistical probabilities of trends and correlations. Usually this sort of calculations is necessary whenever correlations and trends are calculated in the study and it is not required to specify their use in the title of the article.

I was concerned about the representativeness of a single set of time series to represent the spatial aggregation of such a vast area. I am glad to see that some maps were provided, but the authors should emphasize that although the signal of the region agrees predominantly with many parts of the region, there are still some places where that have a different response (see for example Fig. 9b, where not all the region has positive trends).

How did you deal with the calculation of ET in the SPEI index when Temperature was below zero? Is there any case of that?

Specific comments

L47-48. Change “has been become” to “became”.

L48. Over ALL China?

L54. What other perspective could we been talking about?

L73. Change “(SPI)” to “(SPI; WMO, 2012)”.

L74. Change “(PDSI)” to “(PDSI; Palmer, 1965)”.

L75-76. Change “The PDSI is a climatic water balance” to “The PDSI is based on a primitive water balance”.

L109. Altai is not identified in the map. Label of Kunlun Mountains is difficult to read in the map. Include small map of location with respect to entire China.
L122. It is necessary that Kaidu River and Tianshan Mountains be included in Fig. 1.

L137-138. Change “To quantify drought, SPI and SPEI which have a multiscalar character to determine drought variability, were used.” To “SPI and SPEI were used to quantify drought as their multiscalar character helps to determine drought variability”.

L140-141. Change “Thornthwaite” for “Thornthwaite (1948)” and remove “(1948)” from L141.

L142. In Figure 3, we also see time scales of 36 and 48 months. Either include those to the list or remove them from the Fig. 3, as they are not mentioned any more in the analysis. I suggest the latter.

L154. Are those mountains in Fig. 1 also considered arid? Are you referring that you used only the stations in the low-lands?

L155. Add the word “respectively” after “0.25oC/10a”.

L158. What do you mean by highly volatile? I can see that the standard deviations before and after 1997 are similar. If in doubt use a statistical test.

L161. Add “(not shown)” at the end of the paragraph.

L166. Add “(not shown)” at the end of the paragraph.

L167-170. I think this idea is repeated here.

L171. Change “a warmer and wetter” to “a warmer and wetter trend”.

L178. What do you mean by “high consistency”? Change “consistency for most stations” to “consistency between stations” to see if this helps clarify the idea.

L185. Change “More wet events” to “More wet events of less dry events”.

L193-195. AE responds lo energy and water availability. In arid regions, the AET is considered water-limited, as there is ample energy (radiation) available for evaporation.

L199-206. What is the purpose of showing these correlations?

L211-212. But for ET the correlation from the figure seems insignificant.


L215-217. High precipitation will produce more availability for ET, and therefore ET will approximate P (it is still water-limited). That is the reason for the higher correlations of ET-P for higher P.

L229. More positive values of what: SPEI or SPI?

L240. Do you mean SPI here instead of SPEI? SPEI depends on ET.

L242. How about radiation? It is a very important control on PET.

L249. What about the sharp decrease in PET from 1975 to 1995?

L264-274. It is expected for the higher time scales to present lower frequencies and therefore the 12-month is expected to have a higher number of dry months as the high frequency noise is attenuated. Periods where the temperature increase will probably increase this difference. So, I don’t agree with the statement in L272-274 that the difference is a result of previous (multiyear) moisture deficiency.

L326-327. 1.1oC annually for two decades, that implies an increase of 22oC! Please revise.

L349-351. In my experience the correlation between NDVI and Precipitation changes from season to season, and that could be a reason of the poor correlations.

L369-370. I couldn’t see this correlation. What do the numbers inside the figure 16 mean?

Fig. 10. How can ET be negative? Are these standardized values?

Reviewer 2 ·

Basic reporting

no comment

Experimental design

no comment

Validity of the findings

no comment

Additional comments

This research seems interesting to the field of climatic changes in arid regions of Asia. The structure of the manuscript is appropriate, while a better separation of the aspects between the chapters need to be improved.
1. In Introduction, some case studies or related articles on the impacts of drought in Xinjiang should be presented.
2. Numerous studies already concluded that temperature and precipitation experienced significant increase since 1997 and 1987, respectively. So, this should not be accentuated in your ABSTRACT.
3. It’s really hard to understand why the authors listed figure 4? If it is not used, please remove it out.
4. Line 30: please provide full name of SPEI, SPI.
5. Line 81: please provide full name of WMO.
6. Line 89-90: Please clarify this statement of evaporative demand (ET).
7. Line 102: evapotranspiration (ET)?
8. Please clarify this statement of the ET, AE, and PET in your paper.

Reviewer 3 ·

Basic reporting

The goal of the study is to analyze the temporal and spatial changes of drought in Northwest China. For this the authors use two standardized drought indices, the SPI and SPEI at different accumulation time scales and include other sources of information, including satellite and river discharge data. The authors then explore whether these changes are associated to changes in precipitation and temperature through time. The study has the potential to be an important contribution, but the study could be better structured to facilitate its readability, and the different parts could be better connected to fulfill the overall aim. The different steps of the methodology could be better motivated to understand why these are done and used to test your hypothesis. Besides, the title refers to uncertainty and I could not find any uncertainty analysis or discussion in the study. I hope the authors find the comments below constructive.

It seems to me that the overall aim is not clearly stated. If I understand correctly (inferring by the title and the analysis) the authors aim at analyzing the temporal and spatial changes of drought. Aims one and two, as written now, seem to be three specific objectives within that aim. Number one says that the study aims at evaluating the sensitivity of SPI and SPEI to precipitation and evapotranspiration. Perhaps the authors mean to changes in precipitation and evapotranspiration? In addition, it is not clear why is the sensitivity test needed. Note that, if what is sought is to compare the sensitivity of both indices to variations in P and ET, this would require that the two indices being evaluated are based on both P and ET (in this case doing that is not be possible because the SPI is only based on P).

L 141: Thornthwaite reference incomplete, it should be (Thornthwaite, 1948).

Scientific writing and English should be revised. (e.g. L48 'drought has been become', L181: 'these differences was'.

Experimental design

The description of the methods could be improved. For example, which distributions were used to fit the data when calculating SPI and SPEI? Please give the specific details of the correlations that were calculated.
This section seems to include a description of the study area as well. It would be interesting to know how populated is this region, and how common are drought related problems, to further motivate the study.

‘Continuous observed monthly climate variables…’: state which climate variables. It would also be valuable to have some information about data quality, e.g. whether all the variables had data for the period 1961-2015 or not.
L147: ‘This method has been used to investigate the significance of trends in various hydrometeorological factors’. Please state all of the ‘factors’.

Validity of the findings

L177: ‘According to historical records’ … What do you mean with this? Are there independent data (e.g. reports on historical drought events) in this region? If so, it would be good to reference these to show that these two indices are capturing the events.
L188: ‘but from 1966 until 2015, drought events were qualified only using the SPEI…’ This sentence is not clear, do you mean ‘identified’ instead of ‘qualified’ or do you mean that you only calculated the SPEI for this period?
L192-198: this seems to be part of the discussion.
Why do you think that the correlations between SPEI and SPI are stronger in northern Xinjiang than in southern Xinjiang?
Figure 6: caption is not clear. Did you use data from different meteorological observatories? If you used more than the one from CMA, please state it in the materials section. It is not clear what the plot means: why do you plot the temporal Pearson r’s correlations of the SPI and SPEI with mean P and ET values? What is the physical meaning of this? Motivate this analysis.
L209, ‘The correlation between SPI and SPEI is primary [should be primarily] dominated by mean P’: this is not surprising since SPI is only based on precipitation, thus a high correlation with ET could not be expected.
Figure 7: I still have difficulty understanding why to perform at correlation of an index based on only precipitation, with ET.
L211, ‘These results support the perspective that drought sensitivity is related to changes in average P and ET’: This sentence needs to be re-written or discarded.
L231, ‘This was due mainly to the ET method used to calculate the SPEI indices’: How did you come to such conclusion? Did you compare different ET methods? Why do you think that the reason behind the SPEI identifying an increase in drought severity relative to the SPI is a consequence of the method used to calculate the ET and not e.g. the temperature rise?
L251-252: this sentence describes the methodology. Consider moving it to the methodology section.
L273: Can antecedent moisture deficits really result in a meteorological drought?
L289-303: I really like that the authors refer to the impacts of drought in this region for specific events captured by the indices, but these should be referenced (e.g. L 295 ‘It was reported…’ by whom?).

Additional comments

Iit is not clear in the abstract what is referring to background for the study and what is referring to the results of the study. It starts discussing about the changes in temperature and precipitation, but this information is part of the results (trend analysis)? In order to get an overview of your study, it would be easier for the reader to have a summary that follows a logical order, that is: one or two sentences presenting the background of the problem, the aim of the study (which I could not find in the abstract), the methods and results/conclusions.

---

## Round 0.2 · Major Revisions

· Academic Editor

Major Revisions

Please check the reviewers' comments carefully and make changes accordingly. Once you've re-submitted, your manuscript will likely be reviewed by the same reviewers.

Reviewer 1 ·

Basic reporting

The basic reporting is fine. There are sufficient references.

Experimental design

It is good, but sometimes the objectives of performing some of the analyses are not clear

Validity of the findings

This is fine.

Additional comments

Second Review of:
Spatial-temporal variation of drought in Xijiang, Northwest China during 1961-2015

By:
Yao et al.

General comments

The authors have made a major effort in correcting some of the issues I found in the previous version of the manuscript and I think this a much more acceptable work. Some issues still remain:

1. It is not clear what happens to PET when T<=0. If by definition PET=0 when T<=0, then establish that as a definition. It is not correct to suggest that i=0 when T<=0 would result in PET=0, as the equation 2 would make really "i" an imaginary number (unless by definition i=0 if T<=0). Still, if you decide that by definition i=0 when T<=0, "I" (the heat index) would not be zero as other months of the year could have T>0. However even if I>0, equation 1 could still result in an imaginary number depending on the exponent "m" as T<0.
2. In the abstract, I would think that NDVI is the Normalized Difference Vegetation Index, so it is not necessary to refer to "vegetation NDVI" as the "V" in the acronym means vegetation. However, the first time you use NDVI you should spell out what it means: Normalized Difference Vegetation Index.
3. The first time it is mentioned that Xinjiang is in the western part of China is enough information for the reader. It is not necessary to remind the reader that we are reading a study from a region in China (see lines 65, 178, 184, 185).
4. I am still not sure the objective of the correlations shown in Figure 7. Please mention, why they are important. Establish if they are statistically significant.
5. Results are presented in the discussion section by making references to analyses in Figures not previously presented in the results. The first time the reader is directed to an analysis should be in the results section. (See for example lines 366 and 379).
6. In lines 418-422 and in the figure of soil moisture, the results that imply reductions in streamflow and soil moisture are only valid for the latest period of the record. In most of the streamflow longer record the trend is actually positive. The cause is mainly the increase in precipitation and not the increase in snowmelt, as the time series of Runoff/Precip does not show significant trends. The effect of earlier snowmelt is a seasonal effect, that may or may not influence the annual totals (depending of the seasonality of Precip and PET and their changes under increasing warming conditions).

Reviewer 2 ·

Basic reporting

This study analyses drought evolution in Xinjiang, northwestern China from 1961 to 2015. This study has also provided evidence for increasing drought severity caused by temperature rise and has discussed the effects on vegetation and water resources. The study topic is interesting and the general structure of the manuscript is suitable. The paper is clear and unambiguous, professional English is used throughout.

Experimental design

This is an interesting manuscript on drought trends in northwest China (Xinjiang) and such deserves scientific merit. It falls within the scope of PeerJ. Generally, the logical of paper is good, and the overall of this manuscript is better than the previous version. The description of the methods is sufficient that can clarify the readers.

Validity of the findings

The manuscript has been greatly improved, and I believe that the paper is now ready for publication and only a slight improvement would be to add 2-3 more lines in the Introduction regarding the international context of this study: i.e. could these indices be applied to other countries and continents? Furthermore, there are several aspects authors should carefully reconsider before the paper go publication, and I suggest that a minor revision is required before the paper go publication.
Line 33: what do you mean by “urgent”?
Line 87: Define what is P-ET. Please provide the complete description of the acronyms.
Line 108: The area of Xinjiang is not correct (may be ten times of the actual area).
Line 137: Please, provide the full description of the acronym “GIMMS-NDVI3g”.
Line 201: “until” should be “to”.
Line 226: “correlations” should be “correlation”.
Line 240: Delete the point at the end of sentence.
Line 242: “1962” should be “1961”?
Line 248: “nouthwestern Xinjiang”? It should be northwestern Xinjiang.
Line 279: Add the comma before the respectively.
Line 285: “1951” should be “1961”?
Line 292: Again, please add the comma before the respectively.
Line 321: Please, provide the full description of the acronym “CMIP5”.

Reviewer 3 ·

Basic reporting

The article has potential to be an important contribution, the authors have addressed some of the comments and that is good, however some important aspects that remained unaddressed. The English and scientific writing were not properly revised as suggested during the first revision. For example, at the first revision it was pointed out that the expression ‘has been become’ (now in line 53) was not correct. The authors changed it to ‘become’ and this still seems incorrect as it should be in past tense.

The text is sometimes difficult to follow. There are sentences that are difficult to understand, for instance in line 61-62: ‘… widespread drying is common because precipitation strongly depends on a few precipitation events’, or line 2010, ‘More wet events of less dry events were identified…’. Among other examples.

Experimental design

The article has potential to be an important contribution, however there are some major aspects that need to be addressed. The most important is that it is not clear what is the scientific contribution or the novelty of the study. What does this study adds to those findings of e.g. Vicente-Serrano (2015) and Fang et al. (2013)? How are the three individual objectives reaching the major aim, which is suggested by the title of the study? In addition, the discussion is not used to link the results from the different analyses and it feels that it is up to the reader to do this. Going more into detail into the methodological design, the purpose of the sensitivity evaluation is not clear and there is no description of how this evaluation was done in the Methods section. Some aspects about the calculation of the SPI and SPEI were not included (e.g. an aspect already asked in the first revision: to which distributions did you fit the precipitation (for SPI) and climatic water balance (SPEI) data? Gamma? ). Finally, part of the description of the methods are in the results section, such as the sentence in line 199-201.

Validity of the findings

I liked that the authors find the role of temperature on drought monitoring to be important, this seems to be a new and important finding with respect to previous studies such as that of Li et al. (2016a; 2016b). Ufortunately, the authors could have highlighted more the importance of this novel findings.
Finally, since the large research question is not clearly stated, it is difficult to assess whether this was answered or not.

Additional comments

Given all the later, I think that the manuscript is not ready yet for publication unless it is thoroughly revised.

---

## Round 0.3 · accepted · Accept

· Academic Editor

Accept

This round of review has deemed your revision acceptable and great improvement from the previous version. Based on the reviewers' comments, I am happy to recommend acceptance of your manuscript. Congratulations and please keep PeerJ in mind for your next research.

Reviewer 2 ·

Basic reporting

The basic reporting is good. The study topic is interesting and the general structure of the manuscript is suitable.

Experimental design

It is fine.

Validity of the findings

The manuscript has been greatly improved, and I believe that the paper is now ready for publication.

Additional comments

I am happy with the current manuscript.